# Caesarean section or vaginal delivery for low-risk pregnancy? Helping women make an informed choice in low- and middle-income countries

**Alexandre Dumont**[1]*, **Myriam de Loenzien**[1], **Hung Mac Quo Nhu**[2], **Marylène Dugas**[3], **Charles Kabore**[4], **Pisake Lumbiganon**[5], **Maria Regina Torloni**[6], **Celina Gialdini**[7], **Guillermo Carroli**[7], **Claudia Hanson**[8,9], **Ana Pilar Betrán**[10], **On behalf of the QUALI-DEC consortium**[¶]

1 Research Institute for Sustainable Development, Paris University, Paris, France, 2 Pham Ngoc Thach University, Ho Chi Minh City, Vietnam, 3 Interdisciplinary Chair in Health and Social Services for Rural Populations, Université du Québec à Rimouski, Rimouski, QC, Canada, 4 Research Institute of Health Sciences, Ouagadougou, Burkina Faso, 5 Department of Obstetrics and Gynaecology, Faculty of Medicine, Khon Kaen University, Khon Kaen, Thailand, 6 Evidence Based Healthcare Post-Graduate Program, São Paulo Federal University, São Paulo, Brazil, 7 Centro Rosarino de Estudios Perinatales (CREP), Rosario, Argentina, 8 Department of Global Public Health, Karolinska Institutet, Solna, Sweden, 9 London School of Hygiene and Tropical Medicine, London, United Kingdom, 10 UNDP/UNFPA/UNICEF/WHO/World Bank Special Program of Research, Development and Research Training in Human Reproduction (HRP), Department of Sexual and Reproductive Health and Research, World Health Organization, Genève, Switzerland

¶ Membership of the QUALI-DEC consortium is listed in the Acknowledgments.
* alexandre.dumont@ird.fr

**Data Availability Statement:** The Decision-Analysis-Tool is registered in the Decision Aid Library Inventory (DALI) of the Ottawa Hospital

## Abstract

Women's fear and uncertainty about vaginal delivery and lack of empowerment in decision-making generate decision conflict and is one of the main determinants of high caesarean section rates in low- and middle-income countries (LMICs). This study aims to develop a decision analysis tool (DAT) to help pregnant women make an informed choice about the planned mode of delivery and to evaluate its acceptability in Vietnam, Thailand, Argentina, and Burkina Faso. The DAT targets low-risk pregnant women with a healthy, singleton foetus, without any medical or obstetric disorder, no previous caesarean scarring, and eligibility for labour trials. We conducted a systematic review to determine the short- and long-term maternal and offspring risks and benefits of planned caesarean section compared to planned vaginal delivery. We carried out individual interviews and focus group discussions with key informants to capture informational needs for decision-making, and to assess the acceptability of the DAT in participating hospitals. The DAT meets 20 of the 22 Patient Decision Aid Standards for decision support. It includes low- to moderate-certainty evidence-based information on the risks and benefits of both modes of birth, and helps pregnant women clarify their personal values. It has been well accepted by women and health care providers. Adaptations have been made in each country to fit the context and to facilitate its implementation in current practice, including the development of an App. DAT is a simple method to improve communication and facilitate shared decision-making for planned modes

Research Institute: Decision Aid #1959 'Caesarean section or vaginal birth: Making an informed choice'; and available in French, English and Spanish on Zenodo: https://doi.org/10.5281/zenodo.6720225. We also created an application called QUALI-DEC available in both the Apple and Google app stores. In accordance with EU-GDPR principles, qualitative data (in-depth interviews) are stored on a secure, password-protected server, accessible only to the QUALI-DEC team.

**Funding:** This work was supported by the European Commission via the EU's Horizon 2020 Research and Innovation Program (grant agreement No. 847567 to AD, MdL, HMQN, CK, PL, GC, CH) and the UNDP/UNFPA/UNICEF/WHO/World Bank Special Programme of Research, Development and Research Training in Human Reproduction (WHO Study A66006) to CK, PL and APB). The funders had no role in study design, data collection and analysis, decision to publish, or preparation of the manuscript.

**Competing interests:** The authors have declared that no competing interests exist.

of birth. It is expected to build trust and foster more effective, satisfactory dialogue between pregnant women and providers. It can be easily adapted and updated as new evidence emerges. We encourage further studies in LMICs to assess the impact of DAT on quality decision-making for the appropriate use of caesarean section in these settings.

## Background

Decision-making in relation to the mode of delivery is increasingly done together with women as opposed to medical personnel deciding alone. Participatory and joint decision-making demands are sufficient and help to understand information and decision aids [1]. While previous childbirth experience can strongly influence this decision, women who have never given birth before are often uncertain about vaginal delivery [2]. Women's belief and cultural factors mainly determine women's preference for a planned mode of delivery [3]. Moreover, a lack of complete and reliable information about the potential risks and benefits of a planned caesarean section compared to a planned vaginal delivery contributes to many misconceptions regarding the pros and cons of both options, and reduces women's ability to make an informed choice [4, 5]. For some women, fear of pain is becoming so widespread globally that preference for caesarean delivery is growing even in the absence of medical indications [3]. However, maternal request for caesarean section may place health professionals in a situation of ethical tension between the duty to promote the safest way to give birth (thus physiological birth in the absence of medical indications), and the need to respect the patient's choice [6]. In any case, anxiety and decisional conflict can emerge from a non-shared medical decision when women's values and expectations are not met [7].

Women's uncertainty about the planned mode of delivery affects not only high-income countries, but also low- and middle-income countries (LMICs) [8, 9]. In some Asian countries, women think that a caesarean section is safer than vaginal delivery for the baby, and that a planned caesarean offers the possibility to schedule the delivery on auspicious birth dates [10, 11]. For women in Latin America and sub-Saharan Africa who prefer vaginal birth, some express that a planned caesarean is a medical decision because they lack reliable information and empowerment in the decision-making process [10–13]. As a consequence, caesarean rates continue to rise in LMICs and, in many cases, to levels well above possible medical needs, resulting in overuse of the procedure and an increase in the risks without clear benefits [14].

According to the Ottawa Decision Support framework, decision aids can improve decision-making by informing patients about the risks and benefits of different options for their health [15]. We developed implementation research to design and evaluate a strategy, called Quality Decision-Making by Women and Providers (QUALI-DEC), to implement interventions targeted simultaneously at women, health care providers, and health systems in order to improve decision-making for planned modes of birth in Argentina, Burkina Faso, Thailand, and Vietnam [16]. The QUALI-DEC strategy combines four active components: (1) opinion leaders to carry out evidence-based clinical guidelines; (2) caesarean audits and feedback to help providers identify potentially avoidable caesarean sections; (3) a decision-analysis tool (DAT) to help women make an informed decision on mode of birth; and (4) companionship during labour to support women during this time, as well as vaginal delivery. We assume that the DAT could enhance women's knowledge on the risks and benefits of both modes of birth, raise providers' awareness about women's attitudes and preferences regarding delivery, facilitate shared

decision-making about mode of birth, and reduce maternal requests for a planned caesarean section in participating hospitals.

The aim of this report is to describe the development of a DAT tailored to the context of LMICs, and to evaluate its acceptability from the perspectives of women and health care providers in QUALI-DEC participating countries.

## Method

### Ethics statement

We received authorisation from the Department of Reproductive Health of the Ministry of Health in Vietnam, and the four participating hospitals ethically approved the research. We obtained ethical clearance for the study from the local and institutional review boards from the Centro Rosarino de Estudios Perinatales of Rosario, Argentina (Record Notice No. 1/20), Khon Kaen University in Thailand, the Ethics Committee for Health Research of Burkina Faso (Decision No. 2020-3-038), the Research Project Review Panel (RP2) in the UNDP/UNFPA/UNICEF/WHO/World Bank Special Programme of Research, Development and Research Training in Human Reproduction (WHO study No. A66006), and the French Research Institute for Sustainable Development (coordinator). For all individual interviews or focus group discussions, formal written consent was obtained from participants.

We designed the DAT using the Ottawa Decision Support framework [7]. We devised a 3-step approach to generate and evaluate the tool. We used data collected in hospitals in Vietnam to identify needs for decision support. Then, we conducted an overview of the literature to provide evidence for decision support, while we assessed acceptability in hospitals in Argentina, Burkina Faso, and Thailand. Table 1 shows the characteristics of the participating hospitals in the four countries.

The target audience of the DAT is low-risk women with a healthy, singleton foetus, without any known medical or obstetric disorders, no previous caesarean section, and eligible for trial of labour at the time of the anatenatal care visits. Women with previous caesarean section, breech or abnormal presentation, twin pregnancy, or any indication for elective caesarean section (pre-labour) are not the target of the DAT because they are at high-risk for caesarean delivery.

**Table 1. Characteristics of the participating hospitals.**

| Characteristic | Vietnam | Burkina Faso | Thailand | Argentina |
| --- | --- | --- | --- | --- |
| | N = 4 | N = 8 | N = 8 | N = 8 |
| Type of hospital | | | | |
| Public without a private ward | 3 | 8 | 0 | 8 |
| Public with a private ward | 1 | 0 | 8 | 0 |
| Private | 0 | 0 | 0 | 0 |
| Level of care | | | | |
| Tertiary | 1 | 2 | 7 | 8 |
| Secondary | 2 | 4 | 1 | 0 |
| Primary | 1 | 2 | 0 | 0 |
| Teaching hospital | | | | |
| Yes | 2 | 3 | 8 | 8 |
| No | 2 | 5 | 0 | 0 |
| Range of annual births | 2800–42000 | 2500–6000 | 2500–7500 | 614–4945 |
| Range of caesarean rates | 23%–54% | 21%–48% | 36%–56% | 30%–45% |

### Step 1: Identifying needs for improved decision-making on mode of birth

We carried out qualitative research in August 2018 and March 2019 in four hospitals in Vietnam, purposefully selected by the Ministry of Health to reflect a range of contexts (Table 1).

We held individual interviews and focus group discussions among different key informants, including postpartum women, their relative or companion, and health care providers. Maximum variation sampling was used to achieve a diverse sample of providers, including hospital or service managers, clinicians of different qualification (obstetricians, midwives, nurses), sex and seniority. The same method was used to achieve a diverse sample of postpartum women in terms of age, religion, ethnicity and mode of birth (caesarean or vaginal delivery). We recruited and interviewed women and their companion separately, immediately after delivery and before discharge from the study hospitals. Each interview and focus group was facilitated in the participants' respective languages by a female data collector with experience in conducting in-depth interviews or focus group discussions, and audio-recorded if consent was obtained. We conducted focus group discussions separately with obstetricians and midwives to encourage the expression of opinions outside of the clinical hierarchy. We asked women and providers about the possible reasons for the high rates of caesarean section in Vietnam and what they needed to prepare them to discuss the most appropriate planned mode of delivery. Based on the ecological model to understand factors influencing caesarean rates [17], we analysed the recordings and interpreted the data using a thematic analysis approach.

### Step 2: Providing evidence for a holistic decision support tool

We designed the DAT in two sections (S1 Text). The first section aims to inform women during antenatal care (ANC) visits about the risks and benefits of caesarean section and vaginal delivery. The second section aims to help women clarify their values and thus prepare them to discuss their preferences with a health care professional during following visits.

We performed an overview of the literature to provide evidence-based information on the risks and benefits of both modes of delivery. We included systematic reviews (SRs), overviews, or agency statements/reports that provide risk estimates for short- and long-term maternal and child outcomes of women who planned for a caesarean section (but in few cases may have had vaginal delivery instead) compared to women who planned for a vaginal delivery (but ended up with either a vaginal delivery or an emergency caesarean section). We took this approach to ensure the inclusion of studies that reflect the relevant risks for pregnant women who were planning modes of delivery during the antenatal period. In cases where there was no information about planned modes of delivery, we present the evidence about the comparison between all types of caesarean (elective or intrapartum) and all types of vaginal delivery (planned or not, spontaneous or assisted). We ran the search in MEDLINE on 18 April 2018 and updated it on 30 April 2020 using the terms 'delivery, obstetric/adverse effects'[Mesh]) OR ('Caesarean Section/adverse effects'[Mesh]) and filters (Books and Documents, Meta-Analysis, Review, Systematic Review from 2000–2020). If we identified more than one source of evidence for the same outcome, we used the most recent source document (systematic review or overview or agency report) or the source document with the most recent date of search for its evidence base. We included articles in all languages except in Chinese. The search was complemented by the snowball technique; that is, looking for potentially relevant studies on the same subject going backwards (reviewing citations of the key study) and forwards (identifying articles citing the key study). We assessed the quality of individual observational studies included in the SRs using the Newcastle–Ottawa Scale (NOS) or the Scottish Intercollegiate Guideline Network (SIGN) tools. To analyse the certainty of evidence for each outcome, we used the Grading of Recommendations Assessment, Development, and Evaluation (GRADE) scale.

### Step 3: Evaluating the decision support provided by the DAT

We prepared a first draft of the DAT (a paper booklet) based on the first overview of the literature in 2018 to show it as an example to the women and providers so they would better understand what we were proposing. We used the theoretical framework of acceptability (TFA) to assess the prospective acceptability of the DAT booklet among women and providers [18]. We defined acceptability as the extent to which women and providers considered the DAT to be appropriate based on anticipated cognitive and emotional responses to the booklet.

As part of the formative research of the QUALI-DEC project, we held individual interviews in February 2020 in Burkina Faso and in May 2020 in Thailand. In Argentina, due to the COVID-19 situation, individual interviews were not possible. We held two virtual discussion groups in July 2020, one with obstetricians (Heads of the Obstetric Services of the eight participating hospitals) and another with midwives from those same hospitals. One of the objectives of the formative research was to understand how and why a decision aid might help inform preferences and improve decision-making processes for women and providers. We selected a total of 24 hospitals with high caesarean section rates in the three participating countries to reflect a range of contexts, such as district hospitals, regional or provincial hospitals, private clinics, and tertiary/academic hospitals (Table 1).

Recruitment and sampling of women and providers were conducted as specified under step 1 described before. Interviews were also conducted with pregnant women. In each selected facility, researchers facilitated contact with women during their antenatal care visit. Pregnant women who participated in the interviews identified either their partner or the person who they would prefer as a labour companion to participate in the study. Women facilitated contact with potential participants, and researchers followed up to schedule an interview. Female researchers was on site to facilitate recruitment, and was not be involved in clinical care of the patient.

We first transcribed all digitally recorded, qualitative data verbatim in the original language used for collection. We analysed and interpreted the qualitative data using a thematic analysis approach. We performed the analyses in the local or contextually relevant language under the supervision of a social scientist of the QUALI-DEC research team in each country. The country-level analysis involved a combined inductive and deductive approach to allow themes to emerge naturally from the data while also synthesising themes based on questions in a semi-structured interview guide. We organised the analyses as a stepwise process: Each country prepared a report in English language interpreting country-level analyses, which we then compared between countries. The main findings of this higher-level analysis were shared and triangulated with researchers during several online workshops.

According to our data management plan, qualitative interview transcripts in original language have been made available for QUALI-DEC researchers only. The translation into English of the thematic analysis (country-level report) will be made available to the public at the end of the project.

## Results

### Perceived needs and gaps for quality decision-making

In Vietnam, we interviewed 28 women and 16 health care providers. We also held four focus group discussions with obstetricians and midwives (Table 2).

The main findings highlighted agreement on the overuse of caesarean section and the multifactorial nature of its overuse in Vietnam. Obstetricians claimed organisational gains and explained that defensive medicine promoted caesarean delivery in their context, while the

**Table 2. Key informants for the development (step 1) and assessment of the decision-analysis tool (step 3).**

| | Development (step 1) | Assessment (step 3) | | | |
| --- | --- | --- | --- | --- | --- |
| | Vietnam (Aug. 2018 & March 2019) | Burkina Faso (Feb. 2020) | Thaïland (May 2020) | Argentina (July 2020) | Total |
| **Pregnant women** | - | 22 | 27 | - | 49 |
| **Post-partum women** | 28 | 16 | 25 | - | 69 |
| **Relatives/companions** | - | 14 | 16 | - | 30 |
| **Hospital director and department heads** | 9 | 8 | 8 | - | 25 |
| **Gynaecologists-obstetricians** | 5 + 20* | 10 | 18 | 18* | 71 |
| **Midwifes** | 2 + 30* | 9 | - | 15* | 56 |
| **Nurses** | - | 6 | 33 | - | 39 |
| **Total** | 94 | 85 | 127 | 33 | 339 |

Participating hospitals: Vietnam (n = 4); Burkin Faso (n = 8); Thaïland (n = 8); Argentina (n = 8)

* Number of participants in focus group discussions (FGDs): 3 FGDs with gynaecologists-obstetricians and 4 FGDs with midwives in Vietnam; 1 FGD with gynaecologist-obstetrician and 1 FGD with midwives in Argentina.

women expressed a strong preference for vaginal delivery during their interviews. According to the women, the lack of dialogue between them and health care providers, the lack of preparation for childbirth and pharmacological methods to control pain during labour were identified as strong obstacles to planned vaginal delivery. Women and their caregivers expressed the need to be better informed about the risks and benefits of planned caesarean section and planned vaginal delivery.

## Risks and benefits of both modes of delivery

The literature review identified 984 unique references. After screening the titles and abstracts, we selected 30 records for full text reading and identified an additional 7 references through other sources. At the end of the process, we included 15 documents (mostly SRs) that compare the risks and benefits of both modes of delivery (S1 Fig). The characteristics of the reviews are presented in S1 Table. The majority of the reviews include observational studies (cohort, case–control, or cross-sectional surveys), which were mainly conducted in high-income countries. Only two reviews [19, 20] exclusively include women with low obstetric risk, and two reviews compare planned caesarean section to planned vaginal delivery for short-term outcomes using an 'intention-to-treat' analysis [20, 21]. All other reviews include women who had any type of caesarean section (emergency/elective) or any type of vaginal delivery (planned/actual). Certainty of evidence was moderate for four maternal outcomes (hospital stay, hysterectomy, breastfeeding initiation, complications during future pregnancy) and two infant outcomes (obesity and allergies in adulthood). For the other outcomes, the certainty of evidence was low (Tables 3 and 4).

The advantages of a planned vaginal delivery over a planned caesarean section include a shorter hospital stay, faster recovery, increased chances of starting breastfeeding inmediately after delivery, reduced risks associated with surgery (cardiac arrest), and reduced risk of complications in future pregnancies (uterine rupture, placental abruption, placenta previa or accreta) [19–22, 24]. The disadvantages include possible risk of brachial plexus injury for the baby, increased risk of pain in the perineum and abdomen in the immediate postpartum period, and increased risk of temporary urinary incontinence during the first 2 years after delivery [20–22, 24, 26, 27].

The advantages of a planned caesarean section include less pain in the perineum after delivery and in the first 3 months after delivery, and a reduced risk of urinary incontinence during

**Table 3. Summary of maternal outcomes and certainty of evidence.**

| Outcomes | Clinical evidence | Favours planned VD | Favours planned CS | Certainty of evidence* |
|---|---|---|---|---|
| Hospital stay | Maternal length of stay was significantly longer in women who delivered by planned CS compared to those who had a VD [20, 22] | ▇ | | Moderate |
| Recovery | Women with a CS were more likely to have bodily pain that interfered with their usual activities at 8 weeks and 6 months after delivery than women who had a VD [20] | ▇ | | Low |
| Haemorrhage and hysterectomy | The risk of hysterectomy due to postpartum haemorrhage was two times higher in women having a planned CS versus those with a planned VD [19, 20] | ▇ | | Moderate |
| Risks associated with surgery | A planned CS was associated with a fivefold increased risk of cardiac arrest, compared to planned VD [20] | ▇ | | Low |
| Abdominal/pelvic pain during birth and in immediate postpartum | CS was significantly associated with less pain in the abdomen during labour and delivery and 3 days after [20] | | ▇ | Low |
| Abdominal/pelvic pain in late postpartum | Women who had a planned CS were more likely to report pain in the abdomen at three months after delivery [20–22] and persistant wound pain for 12 or more months [23] | ▇ | | Low |
| Perineal pain | Perineal pain level during birth, 3 days and three months after delivery was significantly lower in women who had a planned CS versus those who had a planned VD [20–22, 24] | | ▇ | Low |
| Breastfeeding initiation | Women who had an elective CS had, on average, a 17% lower chance of successfully starting to breastfeed than women who delivered vaginally [20–22, 25] | ▇ | | Moderate |
| Short-term urinary incontinence | The risk of urinary incontinence was significantly higher in women who delivered vaginally than in those who had an elective CS at 3 months and 1 year after delivery [20–22, 24, 26]** | | ▇ | Low |
| Long-term urinary incontinence | The risk of urinary incontinence was not significantly different after 2 years | ▇ | | Low |
| Complication during future pregnancy | A previous CS (no specification on the type) significantly increased the risk of uterine rupture, placenta praevia, placenta accreta, placenta abruption, miscarriage, and ectopic pregnancy in subsequent pregnancies [20, 22, 24] | ▇ | | Moderate |

VD = vaginal delivery; CS = caesarean section

*Certainty of the evidence was assessed using the Grading of Recommendations Assessment, Development, and Evaluation (GRADE) scale

the first 2 years after delivery [20–22, 26]. The disadvantages include a longer hospital stay, more difficulty in resuming regular life after surgery, more abdominal pain in the first 3 months after birth (including persistent wound pain for 12 or more months), reduced chances of starting to breastfeed after delivery, increased risk of hysterectomy due to postpartum

**Table 4. Summary of infant outcomes and certainty of evidence.**

| Outcomes | Clinical evidence | Favours planned VD | Favours planned CS | Certainty of evidence* |
|---|---|---|---|---|
| Brachial plexus injury | The incidence of brachial plexus injury was lower, with borderline statistical significance, in prelabour CS compared to VD [22, 27] | | ▇ | Low |
| Neonatal cardio-respiratory disorders | Elective CS was associated with a 2-3-fold increase in the risk for neonatal respiratory problems, including transient tachypnea of the newborn, respiratory distress syndrome, and persistent pulmonary hypertension [20, 22, 27, 28] | ▇ | | Low |
| Childhood obesity | There is evidence of a possible association between all types of CS and increased risks for excess adiposity in childhood and adolescence [24, 29] | ▇ | | Moderate |
| Childhood allergies | Significant increased risks for allergic rhinitis, food allergy, and asthma in children delivered through all types of CS compared with children delivered vaginally [24, 30–32] | ▇ | | Moderate |

VD = vaginal delivery; CS = caesarean section

*Certainty of evidence was assessed using the Grading of Recommendations Assessment, Development, and Evaluation (GRADE) scale

haemorrhage, complications during a future pregnancy, and cardiorespiratory complications for the baby and respiratory disorders after birth when delivery is earlier than 39–40 weeks of pregnancy [19–25, 27, 28, 33]. Caesareans are also associated with possible risks of obesity in childhood or adolescence and allergies/asthma later in life [20, 24, 29–32].

There is no or insufficient/conflicting evidence about the risk with caesarean section or vaginal delivery for the following outcomes: (i) for women: thromboembolic disease, major obstetric haemorrhage, postnatal depression, sexuality, faecal incontinence, and infertility; (ii) for the children: admission to the neonatal unit, infection, persistent verbal delay, and infant mortality (up to 1 year).

## Assessing acceptability of decision-analysis tools by women and providers

Table 2 presents the number of participants to the individual interviews and focus group discussions in participating hospitals at step 3. The findings of the qualitative analysis show that most women prefer a vaginal delivery over a caesarean section. However, safety is a concern, and women preferring a caesarean section gave, as a reason for this preference, the feeling of being safer than if having a vaginal delivery. The women reported that a DAT booklet would be very useful to avoid misinformation and misunderstandings, and confirmed the need for comprehensive information. Furthermore, they reported that it would help them to acquire information that they would otherwise have difficulty obtaining from health care professionals, either because they do not dare to ask for it, or because doctors or midwives do not necessarily take the time to inform them during ANC visits. Most women indicated that the last trimester is the appropriate time to provide information on childbirth preparation and delivery methods. The women identified several other sources of information such as family members, social networks or the internet, but they are often pro-caesarean and women consider these sources to be unreliable. Our qualitative analysis also indicates that women would like to be asked about their preference for mode of delivery since they believe they have the right to choose. Women in Thailand and Argentina (according to the midwives) suggested using the DAT in a digital format, especially on a smartphone, whereas in Burkina Faso, the paper format would be more suitable. Women in Burkina Faso who could not read suggested videos, podcasts, or audio messages broadcast on the radio or in the antenatal waiting room.

Among health care professionals, there was consensus in the three countries that strengthening informational spaces for women and actively involving them in decisions during ANC visits and other non-clinical spaces was relevant. They saw the DAT as complementary to other ongoing actions to strengthen communication between health care providers and pregnant women. In Thailand, some doctors did not see the value of using the DAT for women who have already planned to attempt a vaginal delivery, while they acknowledged that the DAT could benefit women who request a planned caesarean section, particularly in the private sector. The Thai health care providers proposed distributing the DAT to pregnant women during childbirth preparation classes. This would allow them to discuss it with their doctor during subsequent visits. Providers in Thailand also recommended that the DAT include information on pain control methods, as very few women have access to epidural anaesthesia in this country. In Argentina, providers suggested the DAT include information on the expected timing and course of labour and birth, and the roles of the health care team and the companion, especially during labour. Based on the interviews with key informants, each country's team proposed adaptations of the DAT to fit with their own context and issued recommendations to facilitate its implementation (Table 5). Communication via social networks was highly recommended in Argentina and Thailand.

### Development process of the decision analysis tool

The first draft of the DAT booklet was written in January 2020 and reviewed by a committee of eight experts of the QUALI-DEC team who were not involved in its development. The second draft of the booklet was available in December 2020 and discussed via video conferences with each country's principal investigators of the QUALI-DEC project in Argentina, Vietnam, Thailand, and Burkina Faso. The adaptations that emerged from the individual interviews (Table 5) were discussed and agreed to fit the DAT to local contexts. The third version of the DAT was registered in the Decision Aid Library Inventory (DALI) of the Ottawa Hospital Research Institute: Decision Aid #1959 'Caesarean section or vaginal birth: Making an informed choice' in September 2021. It meets 20 of the 22 Patient Decision Aid Standards for decision support (S2 Text). The two missed standards include: (i) the chances for maternal and infant outcomes; and (ii) readability levels of the target population. The decision aid is available online on the QUALI-DEC website: www.qualidec.com. As requested by women in some countries, we created an application called QUALI-DEC (Fig 1) available in both the Apple and Google app stores in participating countries: (i) Apple (iOS): https://apps.apple.com/bg/app/quali-dec/id1590535948; (ii) Google (Android): https://play.google.com/store/apps/details?id=com.out2bound.whodat.

## Discussion

Based on the Ottawa Decision Support framework, we developed a decision aid to help low-risk pregnant women in LMICs make an informed choice on their planned mode of birth. The DAT was identified as an unmet need in Vietnam and welcomed by women and health care providers in Argentina, Burkina Faso, and Thailand. Providers recognised the need for tools to better equip pregnant women to participate in discussions and decisions during ANC visits. The DAT can be useful in improving communication between providers and women, and addresses the needs of pregnant women for reliable information about childbirth. While most interviewed women would prefer a vaginal delivery, women appreciate that providers ask about their reasons for choosing a vaginal delivery and discuss the pros and cons of both options with them.

Importantly, the DAT is not intended to replace discussion with health care providers, but to better equip women and provide the basis for more informed dialogue with them [34]. This dialogue is important to build a trusting relationship, which can prevent a non-clinical decision for a caesarean section [34, 35]. In addition, in settings where health care providers have limited time to dedicate to each woman during ANC visits, the DAT can promote a more efficient use of this time.

**Table 5. Recommendations to facilitate DAT implementation.**

| Thailand | Argentina | Burkina Faso |
|---|---|---|
| • Mobile app in addition to the booklet<br>• Flip chart with QR code to access the booklet and the app<br>• Distribute the DAT in ANC parenting school<br>• Include information on the companion's role and labour pain management.<br>• Prmoting the DAT using various social media. | • Paper-based and mobile app should be available (consider social networks)<br>• Include information on women's rights and needs for a positive birth experience<br>• Include as many images (or other visual resources) as possible | • Use the ANC booklet for women who can read<br>• Use videos, podcasts, or audio messages broadcast on the radio or in the prenatal waiting room for women who cannot read |

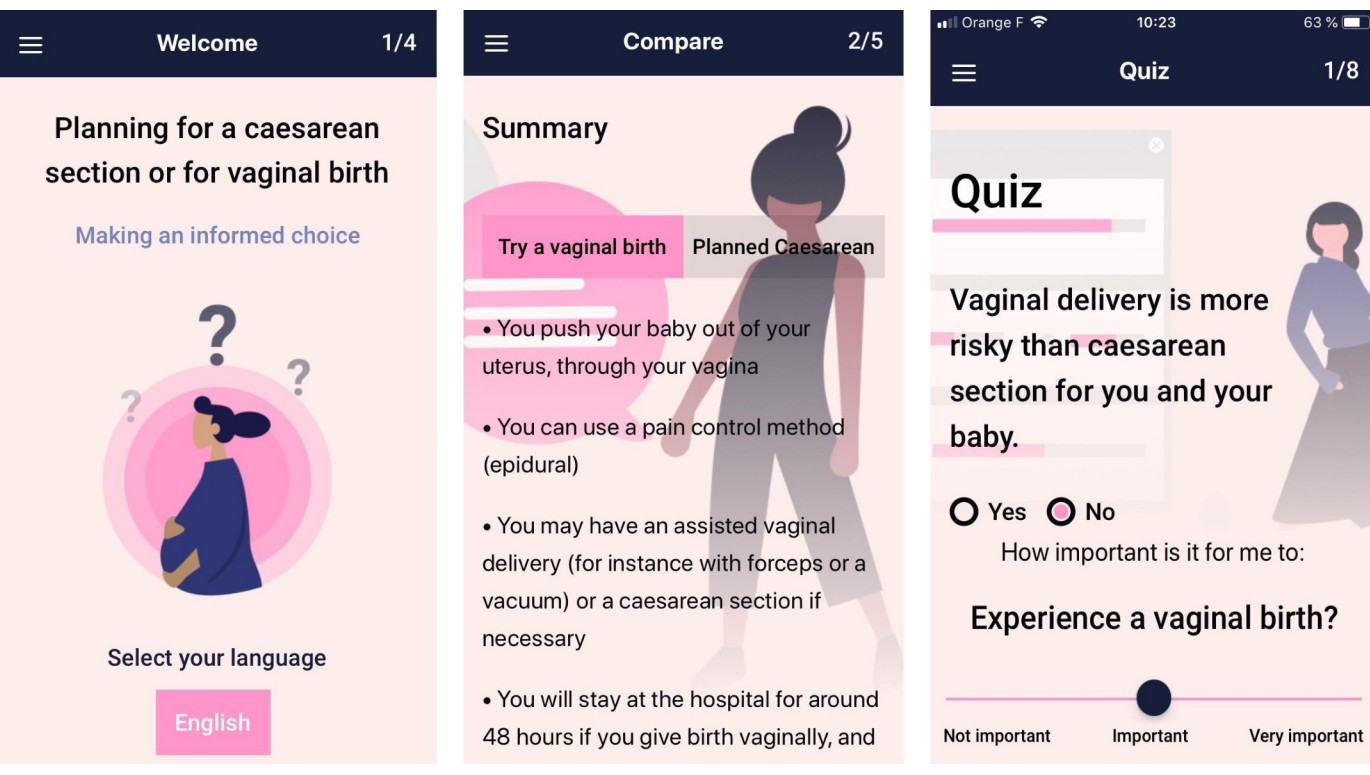

**Fig 1. Decision-analysis tool application for smartphones.**

Our process to develop a decision aid for pregnant women has several strengths. First, our DAT is the first decision aid targeting low-risk women without a previous caesarean section in LMICs. Two studies have previously assessed the impact of a decision aid among women with a prior caesarean section [36, 37], showing improved knowledge among women on the risks and benefits of a trial of labour versus a repeat caesarean section and decreased decisional conflict. Second, the DAT meets 20 of the 22 International Patient Decision Aid Standards for decision support. It includes clinical evidence on the outcomes of both modes of delivery, mainly based on large cohort studies, and helps pregnant women to clarify their personal values. Relevant positive and negative features and outcomes of both options are presented. Finally, health care providers, women, and their companions in various settings have positive perceptions of the tool.

However, we faced some limitations in the development process of the DAT. First, the DAT does not include numerical risk estimates for each of the outcomes according to route of delivery. Because the actual probability of each complication will vary between and within countries depending on multiple factors, we chose not to describe the detailed statistics of each outcome. Instead, we presented, in a single table, the advantages and disadvantages of each mode of birth so that the women would have an overview of the pros and cons of each option in an intelligible, appropriate manner (S1 Text). We felt that this approach is more effective for quality decision-making in LMICs. Indeed, the information given in the DAT was well understood and widely appreciated by the women. Second, the DAT cannot be used by illiterate women. For these populations, we chose to develop other communication media, such as video or audio messages. Third, the clinical evidence on the risks and benefits of both modes of delivery is mainly based on indirect comparisons (e.g., pre-labour or emergency caesarean

section for various maternal and foetal indications versus planned or actual vaginal delivery) and rarely relies on the 'intention-to-treat' approach. Moreover, most of the studies did not adjust for confounding factors that could affect maternal and foetal outcomes such as maternal age, parity, smoking, and body mass index, as well as clinical or obstetric disorders that may have been the primary reason for caesarean section. The inability to disentangle these factors makes it almost impossible to accurately assess risk. Due to the low certainty of the evidence for 8 out of 14 outcomes, the results of our overview of the literature must be interpreted with caution. Moreover, interviews could not be done individually in Argentina due to the COVID-19 pandemic. This undermines the qualitative analysis to assess acceptability of the DAT since the population of that country may be culturally significantly different from the others evaluated. The lack of private hospitals participating in the study is another limitation for findings generalization. Finally, our study provides information about the acceptability of the DAT, but we did not assess the effectiveness or outcomes based on attributes related to the choice made and the decision-making process [15].

Even if causality cannot be confirmed, low-risk pregnant women should be informed that delivery via a planned caesarean section is associated with short- and long-term risks for the mother and the baby, as well as for subsequent pregnancies. The risk of complications with caesarean section, including maternal and perinatal mortality, are probably higher in LMICs than in high-income countries where most of the SRs were conducted, especially regarding risks for future pregnancies [38–41]. For example, the surgical expertise and logistical support (including blood supply) required for a safer caesarean section in cases of abnormal placentation are less likely to be available in low-resource settings [42]. Therefore, women in LMICs could be informed that a planned caesarean section without medical indications can have favourable short-term outcomes, but expose them to potentially severe (and even life-threatening) complications in subsequent pregnancies.

The DAT resonated with the women and health care providers in the four countries of the QUALI-DEC study and is easily adaptable to the specificities of each setting. This tool is designed as a dynamic resource. The booklet and the DAT app can easily be updated as more scientific evidence emerges. If the format is useful and effective, we envisage that important lessons could be drawn on how to implement such a tool in any country or setting.

## Conclusion

Our decision aid for low-risk pregnant women is grounded in the most recent scientific evidence, which does not allow for any doubt about the safety of a planned vaginal delivery compared to a planned caesarean section in low-resource settings. It is expected to build trust and foster more effective, satisfactory dialogue between pregnant women and providers. It can be easily adapted and updated as new evidence emerges. Further studies are needed to improve the quality of the evidence regarding maternal and child outcomes by planned modes of delivery, and to assess the DAT's impact on quality decision-making for appropriate use of caesarean section in LMICs.

## Supporting information

**S1 Table. Characteristics of systematic reviews.**
(ZIP)

**S1 Text. Caesarean section or vaginal birth: Making an informed choice.**
(TIF)

**S2 Text. DALI decision aid information data entry form.**
(ZIP)

**S3 Text. Inclusivity in global research.**
(DOCX)

**S1 Fig. Flow chart of process of study selection.**
(TIF)

## Acknowledgments

We developed this decision-analysis tool using the format of the decision aid created by Marylène Dugas, Nils Chaillet, and Allison Shorten entitled 'Giving birth after a caesarean section: Making an informed choice' (Dugas, 2016) and the recommendations of the research group on the decision support tool from the Research Institute of the Ottawa Hospital (OHRI), affiliated with the University of Ottawa.

## Participating institutions and staff (QUALI-DEC consortium)

Karolinska Institutet (Sweden): Claudia Hanson, Helle Molsted-Alvesson, Kristi Sidney Annerstedt; University College Dublin, National University of Ireland (Ireland): Michael Robson; World Health Organization (Switzerland): Ana Pilar Betrán, Newton Opiyo, Meghan Bohren; Centro Rosario de Estudios Perinatales Asociacion (Argentina): Guillermo Carroli; Liana Campodonico; Celina Gialdini; Berenise Carroli; Gabriela Garcia Camacho; Daniel Giordano; Hugo Gamerro; CEDES (Argentina): Mariana Romero; Khon Kaen University (Thailand): Pisake Lumbiganon, Dittakarn Boriboonhirunsarn, Nampet Jampathong, Kiattisak Kongwattanakul, Ameporn Ratinthorn, Olarik Musigavong; Fundacio Blanquerna (Spain): Ramon Escuriet, Olga Canet; Centre national de recherche scientifique et technologique—Institut de Recherche en sciences de la sante (Burkina Faso): Charles Kabore, Yaya Bocoum Fadima, Simon Tiendrébéogo, Zerbo Roger; Pham Ngoc Thach University of Medicine (Vietnam): Mac Quoc Nhu Hung, Thao Truong, Tran Minh Thien Ngo, Bui Duc Toan, Huynh Nguyen Khanh Trang, Hoang Thi Diem Tuyet; Research Institute for Sustainable Development (France): Alexandre Dumont, Laurence Lombard, Myriam de Loenzien, Marion Ravit, Delia Visan, Karen Zamboni.

## Author Contributions

**Conceptualization:** Alexandre Dumont, Myriam de Loenzien, Marylène Dugas, Guillermo Carroli, Claudia Hanson.

**Data curation:** Alexandre Dumont, Myriam de Loenzien, Hung Mac Quo Nhu, Marylène Dugas, Charles Kabore, Pisake Lumbiganon, Celina Gialdini, Guillermo Carroli.

**Formal analysis:** Alexandre Dumont, Myriam de Loenzien, Hung Mac Quo Nhu, Marylène Dugas, Charles Kabore, Pisake Lumbiganon, Maria Regina Torloni, Celina Gialdini, Guillermo Carroli.

**Funding acquisition:** Alexandre Dumont, Myriam de Loenzien, Hung Mac Quo Nhu, Charles Kabore, Pisake Lumbiganon, Celina Gialdini, Guillermo Carroli, Claudia Hanson, Ana Pilar Betrán.

**Investigation:** Alexandre Dumont, Myriam de Loenzien, Marylène Dugas, Charles Kabore, Pisake Lumbiganon, Celina Gialdini, Guillermo Carroli, Ana Pilar Betrán.

**Methodology:** Alexandre Dumont, Myriam de Loenzien, Marylène Dugas, Guillermo Carroli, Claudia Hanson, Ana Pilar Betrán.

**Project administration:** Alexandre Dumont, Myriam de Loenzien, Hung Mac Quo Nhu, Charles Kabore, Pisake Lumbiganon, Celina Gialdini, Guillermo Carroli, Ana Pilar Betrán.

**Resources:** Alexandre Dumont, Ana Pilar Betrán.

**Software:** Alexandre Dumont, Ana Pilar Betrán.

**Supervision:** Alexandre Dumont, Myriam de Loenzien, Hung Mac Quo Nhu, Charles Kabore, Pisake Lumbiganon, Celina Gialdini, Guillermo Carroli, Claudia Hanson, Ana Pilar Betrán.

**Validation:** Alexandre Dumont, Myriam de Loenzien, Maria Regina Torloni, Ana Pilar Betrán.

**Visualization:** Alexandre Dumont, Ana Pilar Betrán.

**Writing – original draft:** Alexandre Dumont, Myriam de Loenzien.

**Writing – review & editing:** Alexandre Dumont, Myriam de Loenzien, Hung Mac Quo Nhu, Marylène Dugas, Charles Kabore, Pisake Lumbiganon, Maria Regina Torloni, Celina Gialdini, Guillermo Carroli, Claudia Hanson, Ana Pilar Betrán.

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
