## [Decision Letter · Decision Letter 0]

3 Aug 2022

PGPH-D-21-01106

Caesarean section or vaginal delivery for low-risk pregnancy? Helping women make an informed choice in low- and middle-income countries

Dear Dr. Dumont,

Thank you for submitting your manuscript to PLOS Global Public Health. After careful consideration, we feel that it has merit but does not fully meet PLOS Global Public Health’s publication criteria as it currently stands. Therefore, we invite you to submit a revised version of the manuscript that addresses the points raised during the review process.

We look forward to receiving your revised manuscript.

Kind regards,

Bethany Hedt-Gauthier, PhD

Academic Editor

Journal Requirements:

1. Please note that PLOS Global Public Health has specific guidelines on code sharing for submissions in which author-generated code underpins the findings in the manuscript. In these cases, all author-generated code must be made available without restrictions upon publication of the work. Please review our guidelines at https://journals.plos.org/globalpublichealth/s/materials-and-software-sharing#loc-sharing-code and ensure that your code is shared in a way that follows best practice and facilitates reproducibility and reuse.

2. Please include a complete copy of PLOS’ questionnaire on inclusivity in global research in your revised manuscript. Our policy for research in this area aims to improve transparency in the reporting of research performed outside of researchers’ own country or community. The policy applies to researchers who have travelled to a different country to conduct research, research with Indigenous populations or their lands, and research on cultural artefacts. The questionnaire can also be requested at the journal’s discretion for any other submissions, even if these conditions are not met.  Please find more information on the policy and a link to download a blank copy of the questionnaire here: https://journals.plos.org/globalpublichealth/s/best-practices-in-research-reporting. Please upload a completed version of your questionnaire as Supporting Information when you resubmit your manuscript.

3. Please amend your detailed online Financial Disclosure statement. This is published with the article. It must therefore be completed in full sentences and contain the exact wording you wish to be published.

4. Please update your online Competing Interests statement. If you have no competing interests to declare, please state: “The authors have declared that no competing interests exist.”

5. We ask that a manuscript source file is provided at Revision. Please upload your manuscript file as a .doc, .docx, or .rtf.

6. Please provide separate figure files in .tif or .eps format and remove any figures embedded in your manuscript file. Please also ensure that all files are under our size limit of 10MB.

Additional Editor Comments (if provided):

Reviewers' comments:

Reviewer's Responses to Questions

**Comments to the Author**

1. Does this manuscript meet PLOS Global Public Health’s publication criteria? Is the manuscript technically sound, and do the data support the conclusions? The manuscript must describe methodologically and ethically rigorous research with conclusions that are appropriately drawn based on the data presented.

Reviewer #1: Yes

Reviewer #2: Yes

Reviewer #3: Yes

2. Has the statistical analysis been performed appropriately and rigorously?

Reviewer #1: N/A

Reviewer #2: N/A

Reviewer #3: N/A

3. Have the authors made all data underlying the findings in their manuscript fully available (please refer to the Data Availability Statement at the start of the manuscript PDF file)?

Reviewer #1: No

Reviewer #2: No

Reviewer #3: Yes

4. Is the manuscript presented in an intelligible fashion and written in standard English?

Reviewer #1: Yes

Reviewer #2: Yes

Reviewer #3: Yes

5. Review Comments to the Author

Reviewer #1: This manuscript describes the formative research and process used to develop and assess a decision aid tool targeting improving informed decision-making around mode of delivery for low-risk women. This is an important area for research both because of the rising global cesarean delivery rates and the need to provide tools to improve communication and shared-decision-making models between pregnant/intrapartum women and their clinicians.

Overall the manuscript is well presented and clear to understand. There are a few points of clarification and other suggestions for the authors.

Introduction:

The authors can consider pointing out regional differences in preferences and rationale for CS or noting the lack of data from certain regions. For e.g. data from sub-Saharan Africa is quite limited, and that available does not suggest that many women are requesting CS in this region, thus stating that in general twice as many women from LMICs request CS compared to women in high-income countries may be an over generalization.

Methods/Results :

• In both the pre-DAT development interviews noted in Step 1 and the interviews and focus group discussions conducted in Step 3, how were participants selected? The authors note which groups of participants were included but now how they were chosen for interviews.

• For the interviews/focus group discussions held in Step 3, it is a bit unclear which version of the draft DAT participants were asked about - is it the version from January 2020 described in the results? If so it is a bit confusing to have the actually description of the DAT come after the results on the evaluation of the DAT.

• In Table 1: consider including some description/delineation of which interviews were pre-development of the DAT and which were performed after?

• How did the authors determine which of the summary of maternal/neonatal outcomes from the systematic review to include in the DAT. Was everything included or was it based on the GRADE scale result? If some were included in the DAT and not others, I would suggest noting this in Table 3.

• For the assessment of the DAT, did the interviews/ Focus groups address questions the optimal timing of when the DAT should be used i.e., when during antenatal care?

• Given the DAT was developed for use in low-risk women without an indication for CS, is the overall premise of the DAT that generally vaginal birth is considered the route of choice or is it left open-ended with planned vaginal birth vs planned CS as equal contenders? This question speaks to the finding among some providers that perhaps this tool should be used more for women who request a planned CS.

Reviewer #2: Thank you for the opportunity to review the manuscript entitled, “Caesarean section or vaginal delivery for low-risk pregnancy? Helping women make an informed choice in low- and middle-income countries.”

SUMMARY:

The authors have presented the design, process, and evaluation of a decision analysis tool to support women/birthing people’s decision around birth type (vaginal or caesarean). Through a systematic review and qualitative approach in 4 countries, the authors have shared the experience of putting the tool together and its acceptability for implementation. The authors are to be commended for designing a tool to address an unmet need and providing an implementation pathway with both a paper tool and digital app. While I am enthusiastic about the paper and related findings, there are a few comments for authors’ consideration to improve the manuscript, particularly around streamlining and clarity of the messaging.

MAJOR:

• As part of the literature review for the evidence-based information, did the systematic review include international, national, or local guidelines and standards (i.e. World Health Organization guidelines, Ministry of Health guidelines in the 4 countries)? Can authors share more information on in inclusion or exclusion of those guidelines?

• Throughout the paper, it would be helpful to streamline the repetitive writing: for example, Lines 221-225 repeats information that is available in table 2; lines 299-304 repeats information as well. Please consider revising the paper to reduce the repetitions.

• In lines 353-355, the DALI registration and DAT meet 20 out of 22 standards; please add more information about what two standards were missed and if they are addressable.

• As noted above, the clear dissemination of the tool in paper and digital app is to be commended. Can the authors share information about the method of dissemination or ways that the tool is being shared beyond a peer-reviewed publication? How are the authors planning to get the tool in the hands of women and birthing people?

MINOR:

• As noted in the PLoS Global Public Health best practices section on, “Inclusivity in Global Health,” it would be helpful to have the authors share their approach to authorship.

• The abstract starts with the statement, “Women’s fear and uncertainty about vaginal delivery and lack of empowerment in decision-making generate decision conflict and the overuse of cesarean sections (CS) in LMICs.” This framing suggests that it is individual birthing people’s fault that CS rates are high when there are other significant factors at play, including the health system, cultural norms, facility level characteristics. Please consider reframing the background in the abstract or providing more information in the full background section about women’s role or lack thereof in decision making.

• While this is obviously an editorial decision, I found the acronyms of CS (caesarean section) and VD (vaginal delivery) to be halting through the reading process. I am not sure if the word count would be deeply affected, but it would be easier on readers to avoid the acronyms if possible.

• Under the Methods section (line 119-121), the authors state that the women with previous caesarean, breech, or abnormal presentation, or those with elective caesarean are not the target audience; it would be useful to understand “why?” for readers who do not work in maternal and newborn health.

• In lines 131-140, the authors present the goals and populations of the individual interviews and focus group discussions. How did the authors determine the number of interviews to conduct and what were the indicators for achieving saturation?

• In lines 139-140, the authors discussion the analysis of recordings and interpretation of data. Later in Step 3, it is shared that recordings and analysis were conducted in local language and then summarized and cross compared. Were the interviews in Step 1 retained and analyzed in Vietnamese? If translation was completed here (step 1 or step 3) please specify at what stage translation to English was done and by whom.

• Line 163: Any specific reason Chinese articles were not included?

• Table 2: Please consider adding the goal of each round of discussion s if they are different across the countries because the timing is different between the countries.

• Lines 306-308, can the authors describe what is meant by “safety” as a concern for women, or identify what women mean by safer?

• Please provide more information on data accessibility and restrictions.

Reviewer #3: This is a research article on aimed at developing a decision analysis tool (DAT) to help women with a low-risk pregnancy make an informed choice about the mode of delivery and to evaluate it’s acceptability in 3 LMICs. The background, methods and strengths are well described.

It’s unfortunate that due to the COVID-19 pandemic, interviews could not be done individually in Argentina. This undermines the qualitative analysis since the population of that country may be culturally significantly different from the others evaluated. This needs to be acknowledged in the limitations.

It is stated in the article that this tool may benefit women who request a planned cesarean section, particularly in the private sector, however, one of the limitations that this tool may have is that no private hospitals participated in the study. This should also mentioned as a limitation.

I could not locate and download the application in the Apple Store since the app is currently not available in my country or region. This needs to be addressed.

On page 18, line 368, “heath care” should be corrected to “health care”.

Overall is a good, interesting and enjoyable read.

6. PLOS authors have the option to publish the peer review history of their article (what does this mean?). If published, this will include your full peer review and any attached files.

**Do you want your identity to be public for this peer review?** For information about this choice, including consent withdrawal, please see our Privacy Policy.

Reviewer #1: No

Reviewer #2: No

Reviewer #3: No

---

## [Decision Letter · Decision Letter 1]

17 Oct 2022

Caesarean section or vaginal delivery for low-risk pregnancy? Helping women make an informed choice in low- and middle-income countries

PGPH-D-21-01106R1

Dear Dr Dumont,

We are pleased to inform you that your manuscript 'Caesarean section or vaginal delivery for low-risk pregnancy? Helping women make an informed choice in low- and middle-income countries' has been provisionally accepted for publication in PLOS Global Public Health.

Best regards,

Melissa Morgan Medvedev, M.D., Ph.D.

Academic Editor

Reviewer Comments (if any, and for reference):

Reviewer's Responses to Questions

**Comments to the Author**

1. If the authors have adequately addressed your comments raised in a previous round of review and you feel that this manuscript is now acceptable for publication, you may indicate that here to bypass the “Comments to the Author” section, enter your conflict of interest statement in the “Confidential to Editor” section, and submit your "Accept" recommendation.

Reviewer #1: All comments have been addressed

Reviewer #2: All comments have been addressed

2. Does this manuscript meet PLOS Global Public Health’s publication criteria? Is the manuscript technically sound, and do the data support the conclusions? The manuscript must describe methodologically and ethically rigorous research with conclusions that are appropriately drawn based on the data presented.

Reviewer #1: Yes

Reviewer #2: Yes

3. Has the statistical analysis been performed appropriately and rigorously?

Reviewer #1: N/A

Reviewer #2: N/A

4. Have the authors made all data underlying the findings in their manuscript fully available (please refer to the Data Availability Statement at the start of the manuscript PDF file)?

Reviewer #1: Yes

Reviewer #2: No

5. Is the manuscript presented in an intelligible fashion and written in standard English?

Reviewer #1: Yes

Reviewer #2: Yes

6. Review Comments to the Author

Reviewer #1: All comments have been addressed.

Reviewer #2: Thank you for the response to reviewers and modifications to the manuscript. Qualitative data are not available in alignment with the EU-GDPR principles and IRB requirements; this should not be a barrier to publication.

7. PLOS authors have the option to publish the peer review history of their article (what does this mean?). If published, this will include your full peer review and any attached files.

**Do you want your identity to be public for this peer review?** For information about this choice, including consent withdrawal, please see our Privacy Policy.

Reviewer #1: **Yes: **Adeline A. Boatin

Reviewer #2: No
